# Using Machine Learning Approaches to Explore Non-Cognitive Variables Influencing Reading Proficiency in English among Filipino Learners

**Allan B. I. Bernardo** [1,*], **Macario O. Cordel II** [2], **Rochelle Irene G. Lucas** [3], **Jude Michael M. Teves** [2], **Sashmir A. Yap** [2] and **Unisse C. Chua** [2]

1   Department of Psychology, De La Salle University, Manila 1004, Philippines
2   Dr. Andrew L. Tan Data Science Institute, De La Salle University, Manila 1004, Philippines; macario.cordel@dlsu.edu.ph (M.O.C.II); jude.teves@dlsu.edu.ph (J.M.M.T.); sashmir.yap@dlsu.edu.ph (S.A.Y.); unisse.chua@dlsu.edu.ph (U.C.C.)
3   Department of English and Applied Linguistics, De La Salle University, Manila 1004, Philippines; rochelle.lucas@dlsu.edu.ph
*   Correspondence: allan.bernardo@dlsu.edu.ph

**Abstract:** Filipino students ranked last in reading proficiency among all countries/territories in the PISA 2018, with only 19% meeting the minimum (Level 2) standard. It is imperative to understand the range of factors that contribute to low reading proficiency, specifically variables that can be the target of interventions to help students with poor reading proficiency. We used machine learning approaches, specifically binary classification methods, to identify the variables that best predict low (Level 1b and lower) vs. higher (Level 1a or better) reading proficiency using the Philippine PISA data from a nationally representative sample of 15-year-old students. Several binary classification methods were applied, and the best classification model was derived using support vector machines (SVM), with 81.2% average test accuracy. The 20 variables with the highest impact in the model were identified and interpreted using a socioecological perspective of development and learning. These variables included students' home-related resources and socioeconomic constraints, learning motivation and mindsets, classroom reading experiences with teachers, reading self-beliefs, attitudes, and experiences, and social experiences in the school environment. The results were discussed with reference to the need for a systems perspective to addresses poor proficiency, requiring interconnected interventions that go beyond students' classroom reading.

**Keywords:** reading proficiency; non-cognitive variables; machine learning; support vector machines; motivation; growth mindset; reading self-concept; bullying; school connectedness; PISA

## 1. Introduction

Reading literacy is an essential competency for academic learning; high levels of reading proficiency are especially important for higher education, where students are required to access and process information in texts in different domains of learning in school [1–3] and in other aspects of adult life [4,5]. This is partly why international assessments of education have focused on reading as one of the domains to be tested. For example, the Programme for International Student Assessment (PISA) regularly assesses 15-year-old students' reading proficiency together with their science and mathematics proficiency. In the PISA 2018, the Philippines ranked last among 79 countries in reading [6]. Around 80% of Filipino students who participated did not reach a minimum level of proficiency in reading (Level 2); this is one of the largest shares of low performers amongst all PISA-participating countries. The PISA 2018 provides extensive data on a wide range of factors that can be explored to understand students' proficiency in various domains. Previous studies on the performance of Filipino students have inquired into different

aspects of the PISA assessment, such as the alignment of the Philippine reading curriculum with the PISA reading assessment framework [7], school resources and school climate [8], socioeconomic status and students' beliefs [9]. In this study, we use machine learning approaches to explore a wide range of non-cognitive factors that may account for the poor reading proficiency of Filipino learners. The aim was to provide models that can distinguish between Filipino students with low reading proficiency and those with better reading proficiency using different machine learning classification approaches to analyze various non-cognitive factors related to Filipino students' home backgrounds, learning beliefs and motivations, classroom and school experiences, among others.

### 1.1. The PISA 2018 Reading Assessment and Philippine Results

The PISA 2018 framework for reading proficiency features a "typology of cognitive processes involved in purposeful reading activities as they unfold in single or multiple text environments" [10] (p. 36). More specifically, three broad categories of cognitive processes are assessed with more specific cognitive processes specified in each category: (a) locating information (accessing and retrieving information within a text, searching for and selecting relevant texts), (b) understanding (representing literal meaning, integrating and generating differences), and (c) evaluating and reflecting (assessing quality and credibility, reflecting on content and form, detecting and handling conflict).

Proficiency levels were provided to guide the assessment of reading, with Level 2 considered as the minimum proficiency standard. Only 19% of Filipino students attained Level 2 proficiency or better. Among the Filipino students who did not reach the minimum, 15.8% of them were classified as the lowest reading proficiency level (Level 1c) or lower. According to the PISA 2018 report, students who were grouped at Level 1c: "… can understand and affirm the meaning of short, syntactically simple sentences on a literal level, and read for a clear and simple purpose within a limited amount of time. Tasks at this level involve simple vocabulary and syntactic structures" [7] (p. 88). In the case of Filipino students, these reading proficiencies refer to reading in English. The Filipino students grouped into this level can perform only the most basic reading tasks after at least five years of formal instruction in reading in English.

In addition to assessing specific cognitive skills in the domain of reading, the PISA 2018 also underscored the importance of several non-cognitive factors in reading, including the readers' motivations, strategies, practices in different situations, as well as the readers' perceptions regarding their teachers' practices, classroom support, and resources for learning at home and in school [10]. The PISA 2018 also had questionnaires for school heads and parents that inquired into the environment, resources, and various forms of support for students' learning. The Philippines opted not to answer the parent questionnaire but had school heads answer the school questionnaire. Overall, the PISA 2018 assessment provides a wide range of factors related to students and their home and school backgrounds and experiences that could be explored to understand important factors that predict reading proficiency.

### 1.2. Predictors of Reading Proficiency

As is true in most domains of learning, reading proficiency is shaped by the synergistic effects of various personal, instructional, and contextual factors [11]. While there is a strong focus on teaching methods and activities in the reading classroom [12,13], these instructional factors are likely to interact with a student's specific dispositions. Attention has been given to learners' general and specific cognitive abilities and intellectual aptitudes, which may constrain their ability to benefit from specific forms of reading instruction [14,15]. However, there are non-cognitive factors such as dispositions and experiences related to reading that also shape reading proficiency. For example, enjoyment of reading [16], range of personal reading activities [17], intrinsic motivation to read [18], reading self-concept [19], and awareness of reading strategies [20], including metacognitive [21] and self-regulation strategies [22] of the reader are all important predictors of acquiring good

reading proficiency. However, other factors that are not specifically related to students' reading experiences are also known to be related to reading proficiency. These factors are typically collectively referred to as motivational factors such as mastery or learning goals [23], task engagement [24], and task persistence [25], but also include factors such as academic emotions [26] and other beliefs such as students' mindsets [27].

Aside from student-related factors, research has also identified contextual factors that influence students' reading proficiency. Contextual factors typically provide resources and support for learning and development processes associated with the acquisition of higher proficiency in reading and other domains of learning. The most pertinent social contexts for students are the home and school environments, with each context involving different actors, social interactions, and resources [28].

Regarding the home environment, several studies point to parents' educational attainment, work status, and home assets as variables that directly affect students' achievement [29,30], and these home assets include cultural and learning resources such as books, art works and music [31]. These different factors tend to support students' efforts and motivations to learn in reading and other domains and are, thus, positively associated with reading proficiency. There is a more complex relationship concerning the availability of ICT resources at home, with some studies suggesting that the purposes of ICT use at home might be a moderating factor [32]. As should be apparent, these factors in the home environment tend to be associated with families' socioeconomic status [32], a factor that is strongly correlated with achievement in the PISA studies [6].

As regards the school environment, we could distinguish between factors in the immediate learning environment of the classroom where students are learning to read, and the broader school environment [28]. Within the reading classroom, the students interact with their teacher and classmates, and the factors that can influence their reading proficiency include the specific pedagogical approaches and learning activities used by the teacher [31], how the teacher provides feedback and support for the students [19,28], and even the teachers' effort, motivations, and enthusiasm for teaching the subject [33]. The social aspects of the classroom climate are also important factors that influence student learning [34], such as whether the classroom fosters either a collaborative or competitive learning environment and nurtures a mastery learning motivation among the students [35].

Beyond the classroom, there are also important factors in the school environment that are known to play a role in supporting student learning and achievement. For example, the resources that each school has for learning, such as information technology [32] and reading materials [33], and also extra-curricular activities to advance students' related skills [36] are shown to be important supports for student achievement in some contexts. Such factors tend to be related to sources of funds and general levels of resources that schools have, referring to the basic infrastructure, materials, and teacher resources [8,33], which are often constrained in developing countries such as the Philippines. Although not related to resources, there are other factors in the school environment that seem to be important as they relate to the social and interpersonal experiences of the student. For example, the school climate [34], the students' social connectedness [37] and exposure to bullying [38] are also found to be significant predictors of achievement in some contexts.

The preceding brief review of some predictors of reading achievement of students is not intended to be a comprehensive summary of all the relevant predictors of reading achievement. Instead, the brief review is intended to provide a sense of the range of factors within the student and arising from students' interactions in relevant social environments, consistent with socioecological [39] and sociocultural [40] models of human development and learning. We also note that most educational research undertakings typically focus on a select number of factors to test specific hypotheses or theoretical models of their relationships with student reading achievement.

### 1.3. The Current Study

The PISA 2018 database provides information on a very wide range of factors that were assessed as possible predictors of students' proficiency in reading, mathematics, and science. Because the 2018 assessment focused specifically on reading, the survey included numerous items and factors that specifically pertained to students' experiences, beliefs, and attitudes related to reading [10]. A few studies have explored predictors of Filipino students' reading proficiency, and these studies focused on a subset of factors considered to be of interest [8,27,41]. In this study, we utilize machine learning approaches to explore a wide range of candidate variables in the PISA 2018 database to predict the reading proficiency of Filipino students. The objective is to understand the factors that relate to the poor reading performance of Filipino students by identifying the most important variables as ascertained by machine learning approaches, particularly the machine learning model that most accurately discriminates poor performing students from the rest of the students.

The specific objective of the study was to identify the key variables from an overall set of 122 variables that could best distinguish the lowest proficiency Filipino students from those that performed around or above the standard. Our primary focus was to distinguish the students who performed significantly below standard according to the PISA reading levels (i.e., Levels 1b, 1c, and below), as these very poor readers are likely to be the ones who will be unable to progress in education and who need to be the focus in educational interventions. Thus, the aim was to identify the variables that best distinguish these poor readers from the rest of the Filipino students, based on the assumption that these variables will point to vulnerabilities in poor readers that could be the target of interventions.

As mentioned earlier, this aim is addressed by first identifying the most accurate machine learning predictor for discriminating very low performing students from the rest of the students. For this purpose, different machine learning classification approaches, particularly binary classification models, were compared to determine the optimal classifier for distinguishing low and better performing students. A binary classification model, during the training phase, uses input data to iteratively tweak the model parameters by minimizing the difference between the model's prediction and the input ground truth label. The stopping condition for the training iterations is typically one of the following: a pre-determined maximum number of iterations is reached, the validation performance is not improving, or the validation performance worsens. These machine learning classification models are evaluated using cross-validation to measure the generalizability of the model and accuracy metrics to measure the prediction performance.

As opposed to the regression model, which finds the best fit curve that predicts the continuous-valued reading performance, a binary classification model searches for a discrete function that maps the input variables to two discrete categories. Previous efforts which used regression models for analyzing PISA reading performance capture only the linear [8,9,42] and quadratic [43] relationships of input variables and the target variable, ignoring their more complex interrelation. Our work utilizes binary classification models, which consider the underlying higher-order relationships between the input variables and the reading level classification of students.

The plan for analysis was guided by previous empirical studies in literacy development and reading education, which indicated the types of candidate variables to be used in the analysis. The variables considered for the analysis could be conceptually organized into two broad categories—personal and contextual variables—with the contextual category further organized into three subcategories—home, classroom, and school variables (see Figure 1). The personal variables refer to beliefs, attitudes, experiences related to reading, and also to motivational variables that apply to learning in general. Home contextual variables refer to characteristics of the parents, socioeconomic status-related variables, and resources for learning at the students' homes. Classroom contextual variables refer to teacher-related variables, including instructional approaches and activities, and to perceived characteristics of the language and reading classroom. Finally, school

contextual variables refer to resource-related variables of the school, other organizational characteristics, and the students' social experiences in their schools.

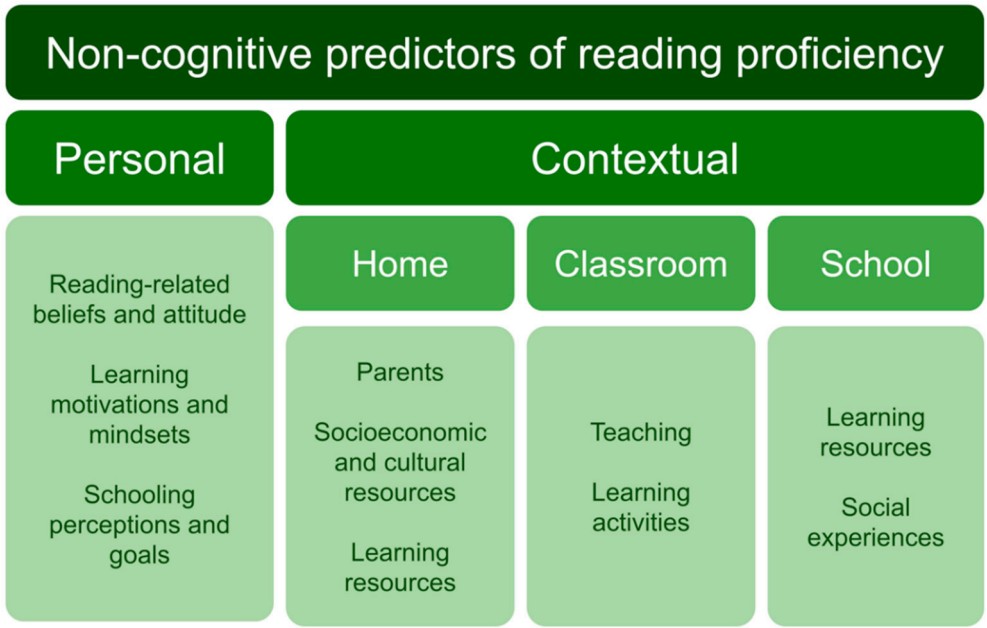

**Figure 1.** Schematic representation of the conceptual framework of the variables in the study.

## 2. Materials and Analytic Methods

### 2.1. The Dataset

The data from the Philippine sample in the OECD PISA 2018 database were used in the study. The data are publicly accessible at https://www.oecd.org/pisa/data/2018 database/, accessed on 17 February 2020. The complete nationally representative sample comprised 7233 15-year-old Filipino students, selected using a two-stage stratified random selection system. Stratified sampling was used to select 187 schools from the country's 17 regions, and then, students were randomly sampled from each school [44].

For machine learning modeling purposes, the students were grouped into the low and high reading proficiency groups. Low-proficiency students are those with poor proficiency at reading levels 1b and below; high-proficiency students are those with reading at levels 1a and better (although their proficiency levels are not actually considered high with reference to the PISA levels). For the students' proficiency levels, we referred to Plausible Value 1 for the reading domain in the PISA dataset. The PISA 2018 assessment does not provide actual reading achievement scores for each student; instead, it measures proficiency in each domain using ten plausible values that represent ten random values drawn from the posterior distribution of the student's scores for reading [6]. We used the first plausible value for the overall reading proficiency; previous studies on the PISA dataset have used only one plausible value [8,9,45] based on the assumption that one plausible value is said to provide unbiased estimates of population parameters. The distribution of students based on their reading level and group is summarized in Figure 2, which also shows that 55% and 45% of the students belonged to the low and high performing groups, respectively.

For the analysis, 122 variables were considered; 41 variables are derived variables or indexes and the rest were single-item responses. Some students had variables with missing values tagged as "M" or "N"; these tags were changed into null values in Python to facilitate data imputation, and for some variables, data were not collected for the Philippine survey. The range of values of each variable was rescaled to 0 to 1. The variables with 100% missing values were dropped from modeling and analysis, and those with a few missing data points were imputed using k-nearest neighbors (kNN). The optimal value for k in kNN was empirically determined by comparing the distribution of the original variable

and imputed variable, using the Mann–Whitney U Test. The $k$ value, which provides the highest number of features that have the same distribution, was chosen; in this case, $k = 7$. After imputation, 90% of the variables followed the same distribution compared to the original.

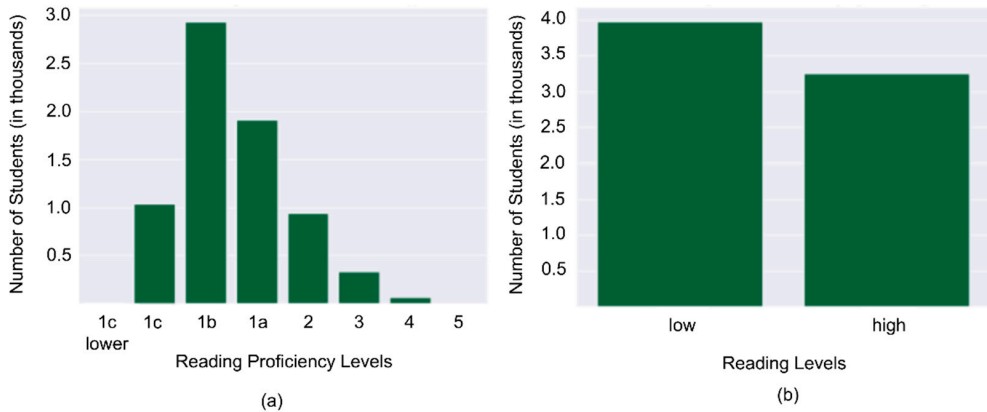

**Figure 2.** Distribution of reading levels of students (**a**) and the distribution of students using high-low groupings (**b**). For machine learning, a comparable distribution of each group, i.e., low and high, is preferable to remove bias in model training.

### 2.2. Machine Learning Modeling

Benchmarking of the machine learning (ML) models was then conducted by optimizing the parameters during training. The dataset was randomly split so that 80% of the samples were used for training the ML models, while the remaining samples were used for testing. The ML models considered were support vector machines (or SVM), Logistic Regression, Multilayer Perceptron, Gradient Boosting Classifier, Random Forest, Ada Boost, and k-nearest neighbors (kNN). Recent work involving the 2018 PISA database [28,45] used SVM-based machine learning approaches to identify high-performing students.

## 3. Results

### 3.1. Machine Learning Modeling Results

The models of these studies achieved average accuracy of, at most, 0.78. Their works are insightful, but we argue that the decision model should be optimal in order for the feature selection to be more valid. The best values for the hyperparameters of each ML model, summarized in Table 1, were chosen through a *grid search* approach. The hyperparameters were tweaked and the values considered are summarized in Table 1. Then, the best performing configuration of the best performing model is used as the final ML model. A summary of the training performance of these models is provided in Figure 3 and a summary of the testing performance is summarized in Figure 4, with SVM as the best performing classifier.

SVM, as the best ML model for this work based on test performance, is a machine learning method that finds a particular linear model by maximizing the space between the decision boundary or hyperplane $z = \mathbf{w}^T\mathbf{x} + b$ and the data points, $\mathbf{x}$, where $\mathbf{w}$ and $b$ are the SVM parameters, $\mathbf{w}$ is the normal vector to the hyperplane, such that the hyperplane margin equals $2/||\mathbf{w}||$ and the offset of the hyperplane from the origin along $\mathbf{w}$ equals $b/||\mathbf{w}||$. By maximizing the space, the SVM increases the total confidence in the prediction. The closest training points to the decision boundary are the support vectors and are used to specify the decision boundary between the classes. Please refer to Figure 5a,b for an illustration of SVM classification on data with two features. For this work, with 114-dimensional data, a radial basis function kernel $k(\mathbf{x}_1, \mathbf{x}_2) = \exp(-\gamma||\mathbf{x}_1 - \mathbf{x}_2||^2)$ is used, with c = 1.0 and $\gamma = 1/(N\sigma)$, where $N = 114$ and $\sigma$ is the variance of $\mathbf{x}$, to transform the

input space into a feature space, such that a hyperplane decision boundary can be found, as illustrated in Figure 5c,d.

**Table 1.** The machine learning models considered for this work (first column), the hyperparameters tweaked for model optimization (second column), and the final values for these hyperparameters (third column).

| Machine Learning Models | Tweaked Hyperparameters | Optimized Value for the Hyperparameters |
|---|---|---|
| SVM | Kernel = polynomial, radial basis function, c = 0.1, 1, 10 | Kernel = radial basis function, c = 1.0 |
| Logistic Regression | c = 0.001, 0.01, 0.1, 10, 100, 1000 | c = 0.01 |
| Multilayer Perceptron | Hidden layers = (32, 32), (32, 32, 16), (32, 32, 32) Activation function = sigmoid, tanh, relu Learning rate = 0.01, 0.001, 0.0001 | Hidden layers = (32, 32, 32) Activation function = sigmoid Learning rate = 0.0001 |
| Gradient Boosting Classifier | n_estimators = 6, 8, 10, 12, 14, 16, 18, 20 | n_estimators = 20 |
| Random Forest | n_estimators = 6, 8, 10, 12, 14, 16, 18, 20 | n_estimators = 20 |
| Ada Boost | n_estimators = 6, 8, 10, 12, 14, 16, 18, 20 | n_estimators = 20 |
| kNN | k = 3, 5, 6 | k = 7 |

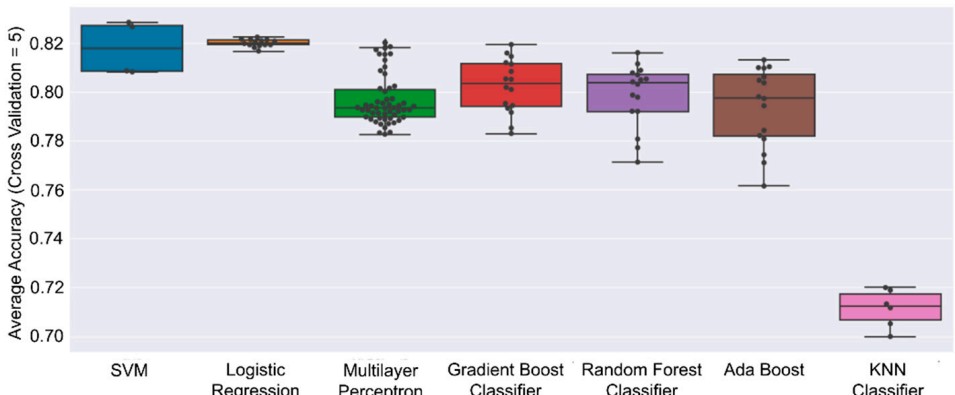

**Figure 3.** Five-fold cross validation training performance of classifiers for different hyperparameter values. The best and worst training performing classifiers are SVM and kNN, respectively.

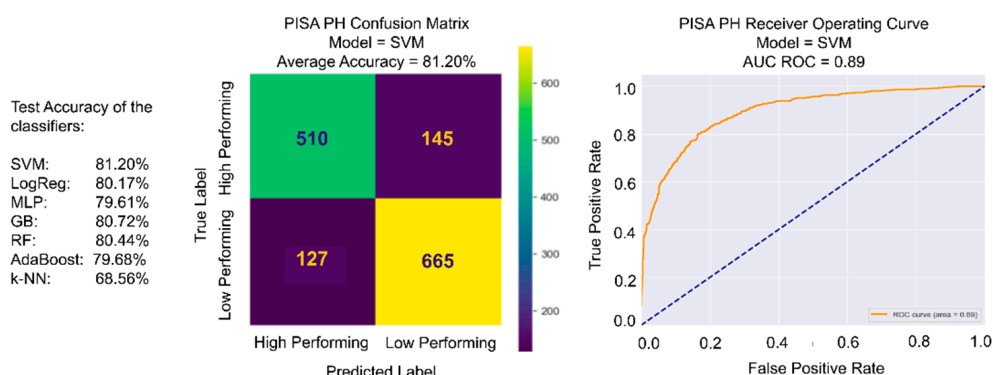

**Figure 4.** Summary of the test accuracies of the classifiers, and the confusion matrix for classifying the low (negative) and high (positive) reading performances using SVM. The best average test accuracy is 81.20% with area under the ROC curve (AUC–ROC) being 0.89. AUC–ROC indicates how separated the two classes are with respect to the trained SVM model. The worst AUC–ROC is 0.5 and the best AUC–ROC is 1.0.

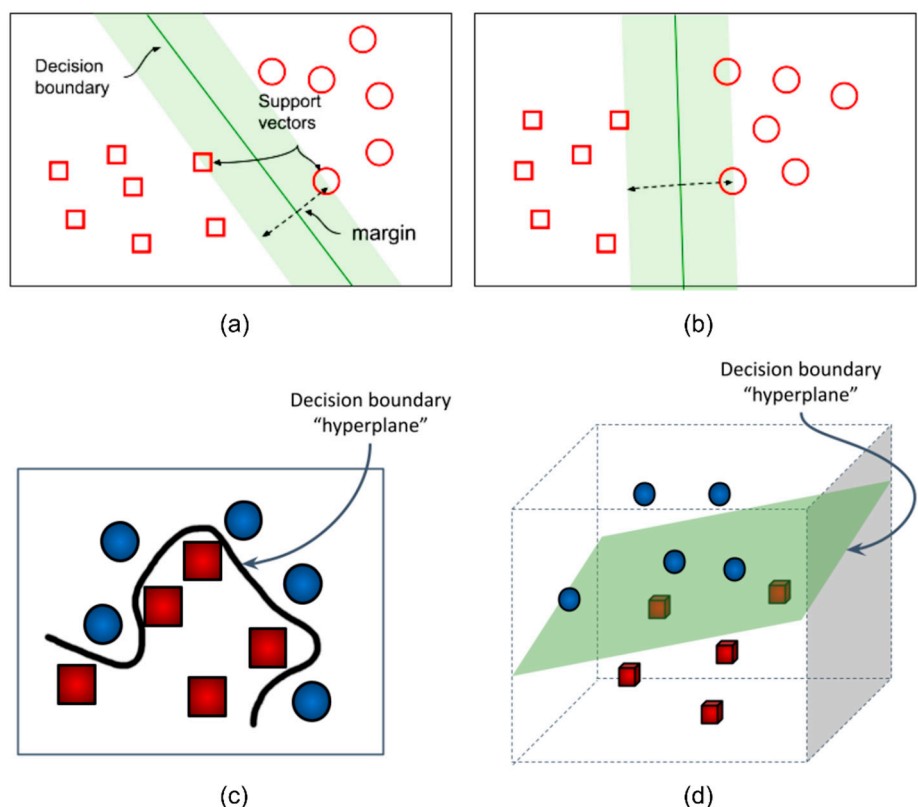

**Figure 5.** (**a**) The SVM decision boundary for a two-feature, i.e., *x*-position and *y*-position, classification (□ or ◯) task, showing the decision boundary, the maximized margin between the two classes of samples, and the support vectors that define the decision boundary. (**b**) A decision boundary that shows that the feature *x*-position is more important than the feature *y*-position in determining the classification of a sample. For a more complex classification task, the input space (**c**) needs to be transformed into a feature space (**d**) via a kernel where it is easier to find a linear model for a decision boundary.

### 3.2. Most Important Variables

The best way to make sense of the SVM model is to take a closer look at the key variables that determine the classification of student performance into low and high categories. For this purpose, we used the SHapley Additive exPlanations (SHAP) [46], which is based on computational game theory. SHAP fairly distributes the gain of each feature by treating each feature as a player in a game. Unlike other interpretability algorithms, it provides a way to compare feature importance at a global level. There are other algorithms (i.e., gradient-based IG) that we could have used for this purpose, and there are more common options such as the SVM-Recursive Feature Elimination (RFE) used in related studies [28,45]. However, unlike SVM-RFE, the Shapley value considers the complete feature set, providing a thorough understanding of the whole logic of the model. SHAP assigns each variable an importance value (*y*-axis) according to their mean absolute SHAP values, and the top 20 variables are summarized in Figure 6. The color bar in each row provides more details regarding how each variable affects reading performance (PV1READ), i.e., positive or negative impact (*x*-axis). Red (blue) dots mean higher (lower) values for a variable.

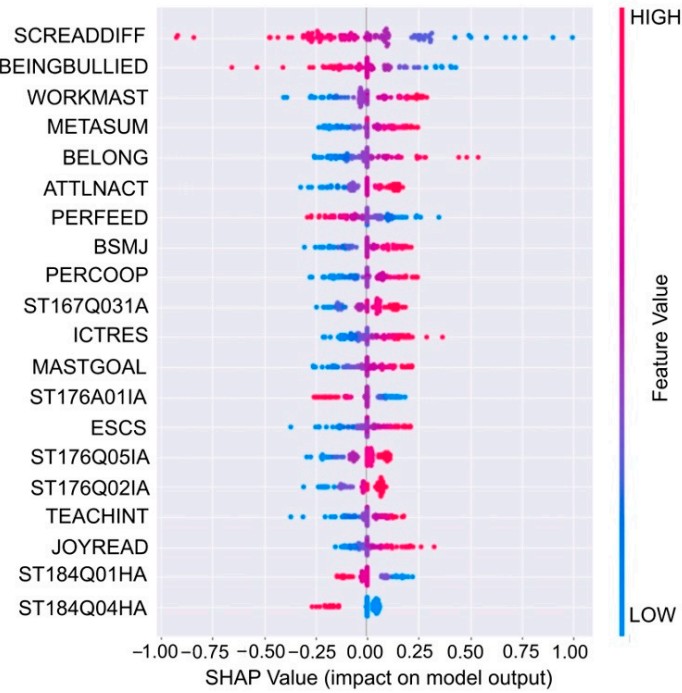

**Figure 6.** Visualization showing the top 20 important variables in descending order. The *x*-axis shows whether the influence of that variable value is linked with higher or lower predictions. Each dot represents a variable value of one training datum. A dot nearer the red color means a high variable value and a dot nearer the blue color means a low variable value.

As illustrated in Figure 6, the most important variable related to the classification of students into low and better performing students in reading is SCREADDIFF. A high SCREADDIFF value has a negative impact on the prediction and a low SCREADDIFF value has a positive impact on the prediction. In other words, SCREADDIFF is negatively correlated with the target variable, PV1READ. Similarly, BEINGBULLIED, the next most important variable, is negatively correlated with the target variable and WORKMAST is positively correlated with the target variable. In summary, of the top 20 variables that are impactful to the prediction of student reading performance, six variables are negatively correlated with the target variable PV1READ, and the rest are positively correlated. Most of the twenty are indexes computed to measure theoretical factors, but five variables are single items that were included as part of some other index and one variable was a single item measure of a factor (i.e., growth/fixed mindset). We discuss these variables with more detail and organize them into meaningful conceptual clusters below.

### 3.2.1. Reading-Related Beliefs and Enjoyment

Four of the top 20 variables are non-cognitive personal variables related to reading. SCREADDIFF is an index computed to measure students' perceived difficulty in reading, and poor readers tended to report higher levels of difficulty. In contrast, poor readers tended to report lower scores for the other three variables: METASUM is an index for the students' metacognitive awareness of strategies for summarizing texts, JOYREAD is an index of students' reading enjoyment, and ST167Q031A is an item indicating that students read fiction because they want to. These four variables indicate that poor readers differ from the better readers in that they have lower intrinsic interest in reading, weaker metacognitive awareness of reading strategies, and more perceived difficulties in reading; these results are consistent with the literature on the role of reading self-concept [19], intrinsic enjoyment of reading [17,18], and metacognitive awareness of strategies in reading [47,48] in students' reading achievement.

### 3.2.2. Teacher-Related/Instructional Variables

There were 3 teacher-related variables among the top 20 variables, 2 of which might be associated with the reading-related variables above. PERFEED is the index on teacher feedback that indicates how often the reading teacher in English tells the student about areas of improvement, and ST153Q04HA is a specific item that refers to a yes–no question about whether their reading teacher in English asks the students to "*give your personal thoughts about the book or chapter*". In both variables, students in the poor reading proficiency group tended to have more positive values, which indicates that their teachers were reported as doing these activities more. These activities are known to be positively associated with students' reading proficiency [19]. However, the result suggests that these might have a negative association with reading proficiency of Filipino students, and we consider possible explanations in the discussion below. The third teacher-related variable is TEACHINT, which is an index of perceived teacher enthusiasm. Students in the poor reading proficiency group tended to report lower values, suggesting that they were more likely to perceive their reading teachers as having low enthusiasm in the classroom. Research suggests that teacher enthusiasm is indirectly related to student achievement in language classes, with teacher enthusiasm directly influencing students' learning engagement in the classroom [49].

### 3.2.3. ICT Resources and Use

ICT-related variables were also important predictors in the SVM categorization model. ICTRES was an index computed to measure the availability of ICT resources in the students' homes, and students in the poor proficiency group were more likely to have low values on this index. However, lack of access to ICT at home is not the only concern, as the poor reading proficiency group also reported low values on being involved in two ICT related activities: ST176Q05IA ("searching information online to learn about a particular topic") and ST176Q021A ("chat online"). Unlike their counterparts who had better reading proficiency, the poor proficiency students were less likely to use ICT for these purposes, which requires reading of texts and presumably supports the learning activities of high school students. These two activities are more active and interactive, compared to the other important ICT related activity, ST176Q01IA ("reading emails"). Poor reading proficiency students were more likely to report higher values on this item. Thus, these students not only have less access to ICT resources at home, but they are also less likely to be involved in using activities that use ICT interactively to support their learning activities; if they use ICT, it is for more passive activities such as reading emails. This pattern of results is consistent with earlier research [50,51].

### 3.2.4. Student Beliefs, Motivations, and Aspirations

Consistent with extensive research on the role of motivational factors in students' reading proficiency [23–25], five motivation-related indexes were found to be high-impact predictors of the SVM model. In four of these variables, students in the poor reading proficiency group reported lower values: WORKMAST, MASTGOAL, ATTLNACT, and BSMJ. WORKMAST is the index computed to represent the motivation and persistence to master given learning tasks, whereas MASTGOAL is the index computed to assess the students' goal of mastery learning. Both indexes emphasize mastery learning as elements of learning motivation across the learning domain, and poor reading proficiency students have low values on both indexes.

ATTLNACT is the index measuring the value of schooling and was measured with items related to the importance of trying hard at school to get a good job or into a good college. Poor proficiency students reported lower values on this index, and relatedly also on the BSMJ index that reports the students' expected occupational status after high school. In a sense, the poor proficiency students have lower pragmatic value for schooling, perhaps because they already set low expectations about the kind of jobs they think they will get after school.

The other motivational variable is a mindset or belief associated with the malleability of their intelligence. ST184Q01HA is a single item that measures agreement with a statement on fixed intelligence; the reverse score of the item is assumed to indicate a measure of the growth mindset. Thus, poor reading proficiency students are more likely to have high values on the idea that their level of intelligence cannot be changed even with effort.

### 3.2.5. Social Experiences in School

Consistent with previous studies also [45,52], three important predictors in the SVM model relate to students' perceptions regarding their social experiences in school. BEING-BULLIED is the index that measures students' exposure to bullying, and the poor reading proficiency students report higher values on this index. However, they report lower values on two other indexes: BELONG in the index measuring the students' sense of belonging in their schools and PERCOOP represents the students' perception that cooperation is encouraged in their school.

### 3.2.6. Economic, Social, and Cultural Status

ESCS is the index computed in the PISA to measure students' socioeconomic status. The measure is derived from student reports on the availability of household items and other possessions and their parents' education and occupational status. The importance of socioeconomic status as a predictor of achievement was observed across almost all countries/territories in PISA 2018 [6], including the Philippines [9,44]. The poor reading proficiency students tended to have low values in ESCS. We can also discern that many of the other important variables are also associated with socioeconomic status, such as the availability of ICT at home, students' learning motivations, and expected occupational status. In the next section, we discuss how socioeconomic status might underlie the most important variables distinguishing low and higher reading proficiency students using socioecological and sociocultural perspectives.

## 4. Discussion

We used several binary classification models to identify the best model to categorize Filipino students as either low or high in reading proficiency and determined that the SVM provided the best model. The top twenty variables (indexes and items) that had the highest impact on the SVM model were identified and these non-cognitive variables characterize the beliefs, motivations, experiences, and resources that distinguish Filipino readers with the lowest proficiency in reading from the rest of the students. Using socioecological [39] and sociocultural [40] theoretical approaches on human development and learning, we can make sense of how the top 20 variables converge in a coherent profile of the poor reading proficiency students in their social environments.

Ecological systems' theory of human development [39] assumes that social interaction processes within a child's social and cultural environments shape all aspects of their development, including their cognitive, emotional, and social cognitive development. These environments range from the most proximal with interactions with parents, siblings, and other family members, to increasing distal environments, such as the classroom and school with interactions with teachers, classmates, and other adults in school. The child's development is even influenced by interactions in more distal environments such as their community and the broader society and its institutions, political and economic systems, social media and others. As regards the development of children's educational and learning-related beliefs, attitudes, motivations, and other psychological functions, we could also see them as being shaped by their interactions in the home, school, and other relevant social environments [53,54]. Even actual educational achievement can be viewed as being distally shaped by these environmental systems [55].

Among the 20 variables that have the strongest impact on the SVM model, ESCS (the index of socioeconomic and culture status) is possibly the best variable that underscores specific characteristics of the social environment that underlie many of the other variables

with a strong impact. Low socioeconomic status (SES) is clearly associated with access to ICTRES (availability of ICT at home), which therefore limits involvement in interactive IT activities (ST176Q05IA and ST176Q02IA) that are helpful in learning. SES has also been shown to be associated with Filipino high school students' motivations. An earlier study on Filipino high school students indicated that SES differences were associated with differences in achievement motivation (including mastery goals), valuing of schooling, and sense of purpose [56]. In that study, students from lower SES environments had lower motivation scores than their counterparts from higher SES environments; this finding echoes the pattern of results found among poor reading proficiency students' values on the motivational variables: WORKMAST, MASTGOAL, ATTLNACT, and BSMJ. More recent research also indicates how the association between fixed/growth mindset (c.f., ST184Q01HA) and achievement was observed only among higher SES students [9]. Thus, the disadvantaged socioeconomic environment of the Filipino student may be associated with several of the highest impact predictors of reading proficiency. Education researchers have long documented moderate to strong SES-related achievement gaps [57,58] and the results of our study among Filipino students provide further evidence on the importance of this factor, but also more specific insights into how SES might be constraining important proximal predictors of student achievement in reading, such as their motivations and effective use of ICT for learning.

Many other factors seem to implicate the important role of experiences in the classroom and in the school that relate to the Filipino students' sense of self as a learner. Specific experiences with teachers could be meaningfully associated with specific reading-related student attributes. For example, the teacher's manner of providing feedback (PERFEED) might aggravate students' self-concept in reading (SCREADDIFF). Consistent with previous research [19], the teachers' lack of enthusiasm (TEACHINT) might reinforce students' own lack of intrinsic interest in reading as an activity (JOYREAD, ST167Q031A). The teacher's manner of engaging students in the reading task (ST153Q04HA) could be a factor for why the students do not have good metacognitive awareness of reading strategies (METASUM). These associations that we propose to give meaning to the results of the SVM model are very consistent with research on reading in other countries [19,20], and align with the socioecological development assumption that interactions in the classroom also play an important role in learners' development—in this case, in the development of reading-related self-beliefs and strategies.

The students' school as a social environment also gives rise to specific experiences that seem to negatively relate to students' reading proficiency. Two of these factors were actually included in the PISA 2018 survey as part of the assessment of student well-being (BELONG and BEINGBULLIED). However, our results show that these aspects of the students' well-being are also associated with their reading proficiency. Collectively, the three variables related to students' school experiences (the third is PERCOOP) characterize the poor reading proficiency students as being socially disconnected from the school; they have a low sense of belonging, perceive their fellow students not to value cooperation, and have frequently been exposed to bullying, a finding consistent with previous research [59]. Feeling socially disconnected in school is most likely going to limit the influence of the school as a social environment for socializing and developing important cognitive, affective, and social goals for these students [60,61], which might explain why these variables have a strong impact on the model predicting reading proficiency.

While many of the variables identified in the SVM model and discussed in this section have been identified as important predictors of reading proficiency and academic achievement in previous studies, there is added insight from the use of the machine learning binary classification models as it revealed a set of variables that have higher order relationships and reading proficiency. These higher order relationships should not be seen as mere mathematical relationships but as representing meaningfully interacting variables that can be understood with reference to models that assume how students' learning and development are shaped by social interaction in different levels of social ecologies [39].

In the foregoing discussion, we highlighted how specific variables seem to arise from the social ecologies of the home, the reading classroom, and the school. However, it is also very likely that there are also meaningful relationships across these environments. For example, the students' SES is sometimes characterized as an important factor in understanding social disconnectedness in school [62], while these feelings of disconnectedness might contribute to lower motivations and values for schooling, as well [63]. The machine learning approach and the socioecological perspective of development help us understand that these variables are working as a system to characterize the attributes and experiences of Filipino students who are reading at very low levels of proficiency.

By implication, attempts to understand how to help Filipino students achieve higher levels of proficiency should also adopt a systems perspective. Helping students with poor reading proficiency cannot simply involve improving curriculum and pedagogy. Instead, it requires multiple interventions and approaches that try to target different variables within the students' various interlinked social environments.

The salient role of the students' families' SES in relation to reading proficiency and to other important variables such as IT resource access and utilization, student motivations, learning-related beliefs and aspirations shows that the problem of low reading proficiency is, to a significant extent, also a problem of poverty. As such, educational improvement efforts need to be embedded in broader efforts to improve the economic conditions of families and communities. However, as a more practical point, these findings point out that students who are most at risk of poor reading proficiency are those students from lower SES families. The results of the study also point to other markers of those at risk of low proficiency, and as such, provide useful guides for targeted interventions in schools. The educational psychology literature points to numerous viable classroom-based or school-based interventions to improve students' mastery-oriented motivations [64,65], beliefs and mindsets [66,67]. Some of these interventions even seem to moderate SES-related achievement gaps [68] and gaps related to students' educational aspirations [69,70].

As the results also suggest a higher-order relationship among specific teaching characteristics, students' reading-related beliefs and strategies, and reading achievement, specific focus could be given to the teaching of reading and the motivation of teachers of reading directed at developing better pedagogical and assessment approaches that will nurture better intrinsic enjoyment of reading and appreciation of effective strategies of reading [71,72]. As regards the social experiences of students with poor reading proficiency, the research literature points to different school-based programs to address bullying [73,74] and to improve social connectedness and cooperation to foster more positive school climates [75,76]. As these approaches suggested in the research literature were developed and studied in other countries, they will need to be contextualized in the Philippine educational communities. Further sustaining the preceding point regarding the students who are known to be at risk and their teachers should be prioritized in such intervention programs.

In conclusion, the study was undertaken as a broader exploration of a wider range of non-cognitive variables that might help characterize the experiences and attributes of Filipino students who were assessed as having poor reading proficiency. Using binary classification machine learning approaches, an SVM model was found to have the best prediction accuracy, and the 20 variables with the strongest impact in the model were meaningfully interpreted as reflecting students' experiences in the home, classroom, and school environment. We acknowledge that our results and inferences are based on our analysis of secondary data, but the quality and breadth of the data provide rich information for exploring factors that predict reading proficiency in the Philippine context. However, our inferences could be tested further and with more confidence, with new data gathered on the same variables assessed specifically to test more precise hypotheses and models. Nevertheless, the results point not only to targets for interventions to help these students improve their reading proficiency but also highlight the need for a systemic view of the students' vulnerabilities and a systemic approach to addressing these students' interconnected concerns.

**Author Contributions:** Conceptualization, A.B.I.B., M.O.C.II, R.I.G.L.; machine learning methodology, M.O.C.II; machine learning modelling and evaluation: J.M.M.T.; data preprocessing and feature engineering, S.A.Y.; data visualization, U.C.C.; writing—original draft preparation, review, and editing, A.B.I.B., M.O.C.II, R.I.G.L.; project administration, M.O.C.II, R.I.G.L.; funding acquisition, R.I.G.L., A.B.I.B., M.O.C.II. All authors have read and agreed to the published version of the manuscript.

**Funding:** This research was funded by grant to the third author from the De La Salle University-Angelo King Institute for Economic and Business Studies (AKI Research Grants 2020–2021 Project No. 500-138), and a Research Fellowship to the first author from the National Academy of Science and Technology, Philippines.

**Institutional Review Board Statement:** The study involved secondary analyses of the officially published PISA 2018 dataset. This dataset was downloaded as a public use file from the OECD website (https://www.oecd.org/pisa/data/2018database/, accessed on 17 February 2020).

**Informed Consent Statement:** Not applicable.

**Data Availability Statement:** The data analyzed in this study are available in the PISA 2018 Database page on the website of the Organisation for Economic Co-operation and Development at https://www.oecd.org/pisa/data/2018database/, accessed on 17 February 2020.

**Conflicts of Interest:** The authors declare no conflict of interest. The funders had no role in the design of the study; in the collection, analyses, or interpretation of data; in the writing of the manuscript, or in the decision to publish the results.

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
