# Peer review of "Using Machine Learning Approaches to Explore Non-Cognitive Variables Influencing Reading Proficiency in English among Filipino Learners"

_education, doi:10.3390/educsci11100628_

Round 1
Reviewer 1 Report
This manuscript presents an interesting and adequate methodology for the prediction of reading proficiency.
Overall, the paper is very well written, with enough results and supporting data, and might be suitable for its publication in COMPAG. I'm not accustomed to this journal or papers from this field, but I find surprising that many results (like model performance) are presented under Material and Methods, not the Result section.
Apart from this, this paper is one of the few and rare cases in which its quality is already enough for its publication in this journal. Machine learning models are correctly developed, validated, interpreted and discussed, and the dataset has been correctly processed.
Congratulations.
Author Response
1. This manuscript presents an interesting and adequate methodology for the prediction of reading proficiency. Overall, the paper is very well written, with enough results and supporting data, and might be suitable for its publication in COMPAG.
Response: We thank the reviewer for these positive comments.
2. I’m not accustomed to this journal or papers from this field, but I find surprising that many results (like model performance) are presented under Material and Methods, not the Result section.
Response: We revised the section so that the plan for analysis was retained as part of the Methods section, and the results related to model performance was moved to the Results section. Please see lines 252 to 270.
3. Apart from this, this paper is one of the few and rare cases in which its quality is already enough for its publication in this journal. Machine learning models are correctly developed, validated, interpreted and discussed, and the dataset has been correctly processed. Congratulations.
Response: We thank the reviewer for these positive comments.
Reviewer 2 Report
Overall: What exactly is the aim of this study? Predicting or understanding or both? The goal, the research question and especially the research gap can be defined more precisely.
The choice of methods follows the aim of the work. In the present work, however, there is no clear separation between understanding and forecasting. In my eyes, these are two completely different tasks. The most accurate forecast does not have to be interpretable. However, in this work, the analysis of the variables follows the most accurate forecast. In my opinion, this is not the optimal way. For even if a decision tree or a regression, for example, do not provide the best forecast, they are much better to interpret than a neural network.
Major:
line 297-300: in order to understand the reading skills, an analysis is needed. SHAP was chosen for this. But why? Why is a SHAP in this context superior? In my oppinion a Shapley-value-based method is required for non-differentiable model types. Why not a gradiant-based explanation method like information gain? Furthermore: Would a regression analysis show different results? Provocative question: What is the benefit of the machine learning approach?
Minor & notes:
line 220 - 230 good description of why poor reading skills of 1a are considered "good" in this context. Good selection of classes.
line 246 and 409: if the goal is the most accurate prognosis possible, an ensamble should be considered. Especially when the methods are so close.
line 259: is the accurace the right performance indicator? What is about the recall and precision? Consider ROC AUC or F1.
line 265: all classifier are in an one percentpoint range of accuracy. What is about the knn? Looks strange. In my experience neural networks performs as good as all other predictors +-3 percentpoints, but not 12.
line 331-348: great insights and questions
Author Response
1. Unclear objective. Prediction or Understanding?
Response: We thank the reviewer for this important question. To clarify, the goal of this study is understanding/explaining the salient variables of the most accurate machine learning predictor that discriminates very low performing students from the others. We have improved our manuscript to reflect this point more explicitly. Please see lines: 162 to 165 and 176 to 178.
2. Most accurate forecast does not have to be interpretable.
Response: Indeed, there are machine learning models that are much easier to interpret. However, we respectfully argue that the decision model to study should be the most accurate model in order for the feature importance study to be more valid.
While there is a known trade-off between interpretability and accuracy, recent advances in algorithms for machine learning model's interpretability and explainability, in our opinion, remove this dilemma. One of these algorithms is the Shapley value used in this work. Shapley value, which is based on computational game theory, fairly distributes the gain of each feature by treating each feature as a player in a game. Unlike other interpretability algorithms, it provides a way to compare the feature importance at a global level. We included a statement in our revised manuscript to reflect this explanation. Please see lines 314-321.
3. line 297-300:
- Why SHAP?
- Why is a SHAP in this context superior?
- Shapley-value-based method is required for non-differentiable model types. Why not a gradient-based explanation method like IG?
Response: Although gradient-based IG requires that model to be differentiable, Shapley value, to our knowledge, makes no assumption on the differentiability of the model. While linear SVM is gradient-based, the non-linear SVM implementation that we used is non-gradient based (thus, differentiability is not guaranteed). We used a kernel trick which forces us to use a non-gradient based approach, particularly, sequential minimal optimization. There are other more common options such as the SVM-Recursive Feature Elimination (RFE). But unlike SVM-RFE, Shapley value considers the complete feature set providing a thorough understanding of the whole logic of the model. We incorporated this in our revised document. Please see lines 316-321.
4. line 220 - 230 good description of why poor reading skills of 1a are considered "good" in this context. Good selection of classes.
Response: Thank you for this positive comment.
5. line 246 and 409 if the goal is the most accurate prognosis possible, an ensemble should be considered. Especially when the methods are so close.
Response: Thank you for the suggestion. However, we are also limited by our current available computing devices. But we will consider this in our future work to have better prediction accuracy.
6. line 259: is the accuracy the right performance indicator? What about the recall and precision? Consider ROC AUC or F1.
Response: Thank you for the suggestion. The main metric to determine the best classification model for a task depends on the application. For example, in identifying whether a sample has a disease or not, preference is on the false negative rate metric rather than the accuracy. For a dataset with highly unbalanced samples, such as in anomaly detection, F1-score is checked rather than accuracy. For determining the features that are salient in classifying student performance, the best metric is accuracy.
We did not consider the other metrics because the main objective is the classification of student performance between students with low performance and good performance. Also, the samples are somewhat balanced (55% low performing vs 45% good performing students). Therefore, the metrics that capture the accuracy for imbalance data, which include F1 score, recall, precision, false negative rates, are not needed.
Nevertheless, we included the ROC curve in Figure 4, in the revised document. Please see line 278.
7. line 265: all classifiers are in a one percent point range of accuracy. What is about the knn? Looks strange. In my experience neural networks perform as good as all other predictors +-3 percent points, but not 12.
Response: KNN or the k-nearest neighbors, is a straightforward scheme of classification where a sample is classified based on the majority class of its k nearest neighbors. We included again the “k-nearest neighbors” near KNN in the manuscript to avoid confusion. Please see line 258.
8. line 331-348: great insights and questions
Response: Thank you very much for your positive comments.
Round 2
Reviewer 2 Report
I enjoyed reading your paper. Thank you very much for the professional work.
Point 7: I´m realy sorry. My bad. I work a lot with neural networks which in german are "Künstliche Neurale Netze" or short "KNN". This abbreviation is so familar me, that i didn´t think twice.